# Psycho-oncology practice for cancer patients during the pandemic lockdown in Italy: A qualitative mixed-method study with psychotherapists

Luca Ghirotto[1], Ludovica De Panfilis[2,3], Marta Perin[4]*, Alessandra Miraglia Raineri[5], Francesco De Vincenzo[6], Matías Eduardo Díaz Crescitelli[1], Elisa Rabitti[5,7], Silvia Di Leo[5]

**1** Qualitative Research Unit, Azienda USL-IRCCS di Reggio Emilia, Reggio Emilia, Italy, **2** Department of Medical and Surgical Sciences, Alma Mater Studiorum – University of Bologna, Bologna, Italy, **3** IRCCS Azienda Ospedaliero-Universitaria di Bologna, Bologna, Italy, **4** Legal Medicine and Bioethics, Azienda USL-IRCCS di Reggio Emilia, Reggio Emilia, Italy, **5** Psycho-Oncology Unit, Azienda USL-IRCCS di Reggio Emilia, Reggio Emilia, Italy, **6** Department of Human Sciences, European University of Rome, Rome, Italy, **7** Palliative Care Network, Primary Care Department, Azienda USL-IRCCS di Reggio Emilia, Reggio Emilia, Italy

* marta.perin@ausl.re.it

## Abstract

### Background

At the beginning of the COVID-19 outbreak, psychotherapy practice underwent a drastic reorganization. To enhance knowledge of the challenges healthcare professionals faced during the pandemic, this study explores the experiences and practices of Italian psychotherapists caring for cancer patients during the first phase of the COVID-19 pandemic.

### Method

This mixed-method study consists of a qualitative cross-sectional survey followed by open-ended semi-structured interviews with a subsample of survey respondents. The data were then triangulated to depict better the experience of caring for cancer patients from the psychotherapists' perspective.

### Results

The final dataset included 102 valid responses. Subsequently, one male and 21 females participated in the interview-based study. Qualitative analysis revealed four themes and specific subthemes: 1. patient relationships (the impact of restrictions on family ties, the impact of restrictions within the healthcare environment), 2. clinical practice management (the use of technologies for psychological interventions, timing and continuity of care, changes in the number of requests for psychological

**Data availability statement:** All relevant data are within the manuscript and its Supporting Information files.

**Funding:** This study was partially funded by the Italian Ministry of Health – Ricerca Corrente Annual program 2025. No additional external funding was received for this study.

**Competing interests:** The authors have declared that no competing interests exist.

interventions), 3. emotional aspects (emotions captured in patients, emotions captured in other professionals, the inner world of the psychotherapists), and 4. organizational recognition (investments in psychological support and service coordination, issues related to employment status).

## Conclusion

Our findings provide knowledge of the pandemic's impact on psycho-oncology practice, offering further input for research on innovative tools in psychotherapy and staff support programs and the development of psycho-oncology services that can systematically respond to the multifaceted needs of cancer patients, relatives, and healthcare professionals.

### Strengths and limitations of the study

We utilized a convenience sampling approach, introducing a selection bias intrinsic to anonymous online surveys. Study findings primarily capture the experiences of those who willingly participated in the research.

Most PTs (i.e., psycho-oncologists with a specialty in psychotherapy) addressed their intervention to adult cancer patients, thereby limiting the transferability of our findings to PTs working with the pediatric population.

Moreover, a high proportion of PTs were from the Italian regions in the so-called "red zones," which were profoundly impacted during the initial wave of the pandemic. Despite these limitations, we gathered a wealth of information from a nationwide spectrum of PTs, encompassing diverse psychotherapy approaches and extensive experience in psycho-oncology. Data collected by the survey were further enriched by those gathered through the qualitative individual interviews, providing a substantial understanding of the experiences and practices of PTs during the outbreak.

While they do not originate from a comprehensive representation of PTs, the insights obtained from our study can serve as a valuable resource for understanding the experiences and adjustments made by PTs during public health emergencies.

## 1 Introduction

COVID-19 emerged in Wuhan, China, in December 2019. By March 2020, Italy faced a severe outbreak, prompting a nationwide lockdown until May 4. This led to a significant reorganization of oncology services [1], with psychotherapists prioritizing in-person care only for critical cases [2] and addressing both illness-related distress and fear of the virus [3,4]. In this context, gaining a comprehensive understanding of psychotherapy practices directed at cancer patients may provide a valuable perspective, particularly given Italy's unique challenges during the COVID-19 pandemic as a country severely affected first.

Psycho-oncology has been recognized in Italy since 1985, with the founding of the Italian Society of Psycho-oncology (SIPO). Over 300 psycho-oncology services are active in hospitals and palliative care facilities, where psycho-oncologists are

engaged in clinical, research, and training activities, including staff support within multidisciplinary teams (data are available online at www.siponazionale.it) [5]. As for clinical activity, psychological interventions are addressed to both cancer patients and their relatives and are delivered by PTs (i.e., psycho-oncologists with a specialty in psychotherapy), usually through in-person clinical sessions. The primary aim of psychological interventions revolves around empowering patients and their families to deal with emotional, relational, and existential concerns connected to cancer.

The available research on psychological aspects within the oncology domain during the pandemic predominantly investigates the distress experienced by cancer patients and their family caregivers in response to the changes brought about by the spread of COVID-19, encompassing shifts at both social and healthcare levels [3,6–11]. The few studies performed on PTs focus on their experience with remote working and video consultations [2,12,13] and on the prevalence and characteristics of peri-traumatic distress they experienced [5]. The only contribution broadly exploring the impact of the pandemic on PTs' activities is the cross-sectional qualitative survey performed by Archer and colleagues [14] in the UK, where a meaningful picture of services and professionals' adaptations under the pandemic strains is provided to the readers. However, the primary emphasis of this survey pertains to research and organizational matters. A deeper understanding of how clinical activities in psycho-oncology were implemented during the pandemic could enhance existing knowledge and provide professionals with valuable insights on how to handle similar crises in the future.

This study explores the experiences and practices implemented by PTs who provided care to cancer patients during the initial wave of the COVID-19 pandemic in Italy. As highlighted above, a detailed understanding of the psychotherapy practice targeting cancer patients during the pandemic could enhance our knowledge of the challenges faced by PTs in such an unexpected and unprecedented historical time and provide professionals from other disciplines with insights into lessons learned from measures and strategies employed to cope with these challenges [14–18].

Our initial research question was: how did Italian PTs experience and adapt their practices in providing care to cancer patients during the initial wave of the COVID-19 pandemic?

## 2 Methods

### 2.1 Study design

We implemented a QUAL→qual mixed-methods design [19], consisting of a qualitative cross-sectional survey [20] followed by open-ended semi-structured interviews with a subsample of surveyed PTs. According to Morse and Niehaus, "QUAL" indicates a qualitatively driven study with a qualitative core component; "qual" indicates a qualitative supplemental component, and → suggests that the supplemental component was conducted sequentially after the core component.

Our study involved a two-phase approach: a survey to gather initial data and qualitative interviews with a selected group of participants who had consented to further involvement. The survey collected PTs' experiences through questions informed by the Critical Incident Technique (CIT) [21,22], a research method used to systematically collect and analyze significant events, or "critical incidents," profoundly impacting an individual's experience or behavior in specific situations. Developed by John Flanagan in 1954, CIT involves gathering detailed descriptions of particularly important or challenging events. These incidents are then analyzed to identify patterns, key themes, and insights into how people respond to specific circumstances. CIs in this study refer to situations perceived as significant by PTs caring for cancer patients during the pandemic. In the study's first phase, some PTs agreed to be contacted for further research (by leaving their contact information). In the second phase, these PTs were invited to participate in qualitative interviews. The purpose of these interviews was to explore more deeply the contents of CIs they had reported during the survey and to gain deeper insights into their experiences as PTs during the initial wave of the COVID-19 pandemic in Italy. Findings from both phases were combined through a process known as triangulation. We integrated data from the survey and qualitative interviews to provide a more comprehensive and multi-dimensional understanding of PTs' experiences during the first wave of the COVID-19 pandemic by combining insights from the survey (CIs) and the in-depth qualitative interviews (that explored these

incidents further), triangulation allowed for cross-validation of findings (Carter et al., 2014), thus enhancing the credibility and depth of the study by ensuring that the results are not based on a single data source but are corroborated through different perspectives, leading to a richer and more nuanced representation of the PTs' experiences. In Fig 1, we provide a representation of this process.

**2.1.1 Survey development and pilot.** In the first section of the questionnaire, socio-demographic and professional information were gathered. In the second section, according to the CIT indications [21], we developed three open-ended questions allowing PTs to describe an event (namely the CI) they perceived as positive and an event they perceived as adverse in caring for cancer patients during the first wave of the lockdown. Then, they were asked to report their thoughts and suggestions about limiting or preventing future adverse events. The questions were proposed as follows:

1. Thinking about your clinical practice in these recent weeks of the pandemic, could you tell us about an event that has a particularly POSITIVE impact on you?

2. Thinking about your clinical practice in these recent weeks of the pandemic, could you tell us about an event that has had a particularly NEGATIVE impact on you?

3. Regarding the negative event you mentioned, what changes would you make to prevent such events from happening again in the future or to ensure a different outcome?

In the last section, the surveyed PTs could provide their contact details if they were willing to be interviewed. The survey was piloted with five conveniently selected PTs to enhance comprehensibility and readability. After this pilot, we re-formulated the open-ended questions according to the feedback we received.

**2.1.2 Survey administration and dataset.** Three team researchers (LDP, SDL, and AMR) distributed the survey online from May 11th to 30th, 2020, using Google Form online software. The survey was disseminated among PTs within stakeholder networks (Italian Society of Psycho-oncology, Italian Society of Palliative Care, and the local palliative care association 'Zero K') through mailing lists, newsletters, social media, and web posts. To be eligible for participation, PTs

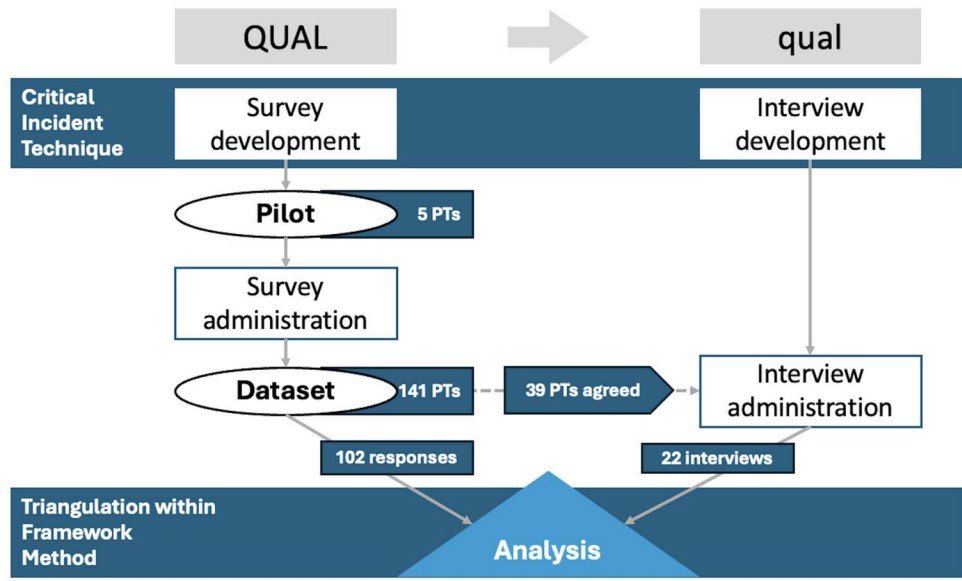

**Fig 1. Mixed-methods design process.**

must have recently been delivering psychotherapy to cancer patients during the pandemic (this was assessed through a triage question in the questionnaire).

**2.1.3 Semi-structured interview development and administration.** We formulated the semi-structured interview guide using the survey questions (Table 1). It encompassed specific prompts designed to explore the CIs reported in the survey by PTs, explore the perceived effects of the current situation on their clinical practice and personal life, and identify the specific concerns, objectives, or tasks that PTs considered necessary while caring for cancer patients.

Three team researchers (LDP, MEDC, and AMR) conducted all the interviews by telephone or remotely from August to September 2020. We audio-recorded and verbatim transcribed the interviews.

## 2.2 Data analysis

Two researchers (AMR and SDL) analyzed socio-demographic and professional information using descriptive statistics. Responses regarding CIs were analyzed using the framework method [23,24]. The framework method systematically analyzes qualitative data, involving several steps: familiarization with the data, developing a coding framework, indexing (or applying the framework), charting, and interpretation. The method is flexible and allows for inductive (emerging from the data) and deductive (based on predefined concepts) analysis. Data are organized into a matrix that enables researchers to visualize patterns and relationships between themes, making it easier to compare responses across participants. The framework method is beneficial for team-based research, as it promotes transparency and reproducibility by structuring how data is analyzed and interpreted. In our study, we used this method to systematically explore the understanding of how the pandemic impacted PTs' experiences and practices through the PTs' accounts and perspectives.

Initially, three researchers (MEDC, MP, FDV) independently analyzed approximately 30% of the PTs' responses to the open-ended survey questions [20], developing an initial coding framework inductively. During this phase, the researchers discussed discrepancies in labeling or identifying recurring concepts in regular meetings. When differences in interpretation arose, the team revisited the original data and deliberated until a consensus was reached. Once the initial framework was established, two additional researchers (LG and SDL) reviewed and refined it, ensuring that the identified themes accurately reflected the data. This step involved further discussions within the team to address any remaining disagreements. This iterative process resolved discrepancies collaboratively, ensuring the final framework was comprehensive and consistent with the data.

Table 1. Interview guide.

| Topic | Exemplifying questions |
|---|---|
| **About the critical incident reported** | • Could you tell me about the incidents you narrated as the most negative during your recent clinical practice?<br>• How did you feel?<br>• How has this impacted your practice? |
| **Clinical practice** | • What is the impact of the pandemic on your practice? Could you please describe how your practice is?<br>• Could you please provide some examples? What are the most impactful issues addressed in this period? Are there any situations that you think are important to share? |
| **Perspectives on patients** | • What are the main concerns of your patients in this pandemic period? |
| **Timing** | • How do you organize your clinical practice?<br>• How has this situation affected the timing in your professional life? |
| **Professional goals and personal life** | • As a psycho-oncologist, how do you interpret your professional goals in this current situation?<br>• What goals do you set for yourself? Could you provide some examples?<br>• How are you, as an individual, experiencing the situation at home? |
| **Closing questions** | • What advice would you like to give to colleagues experiencing the same situation?<br>• Is there anything that came to your mind during the interview that you would like to tell me? |

After these revisions, three researchers (LG, SDL, and LDP) thoroughly checked the second version of the framework before applying it to the entire dataset. During the subsequent analysis of the remaining survey responses, two researchers (MEDC and FDV) used the framework to calculate the frequency and intensity of themes.

For the interview analysis, we used the same framework deductively. Four researchers (MP, ER, MEDC, and FDV) analyzed the responses, again working collaboratively to resolve any discrepancies. This ensured that the framework remained consistent while allowing for in-depth exploration of new insights that emerged during the interviews. All disagreements were resolved through group discussion, carefully ensuring that all voices were heard and that the final coding accurately represented the data.

**2.1.3 Triangulation.** Data triangulation was performed at the data analysis stage. We employed the developed framework to compare CIT and interview data [25]. SDL and LG allocated the interview data into the framework previously defined by CIT. Then, they proposed a final categorization by themes. Therefore, the narrative of the findings considers the analysis of both datasets. All the authors agreed on the results.

**2.1.4 Rigor and reflexivity.** We employed procedures to ensure the reliability of the study. First, at least two researchers followed each study stage (from ideation to reporting). Moreover, coding was shared and validated by the authors through discussion. In this context, having an interdisciplinary team was a positive aspect: the group included a PT, an expert in psycho-oncology and palliative care (SDL), a nurse with expertise in qualitative analysis (MEDC), two philosophers trained in bioethics and qualitative methods (LDP and MP), and three researchers-psychologists (FDV, ER, and AMR) supervised by a qualitative methodologist (LG).

**2.1.5 Ethical considerations.** Formal ethical approval was granted by the AVEN Ethics Committee (in-house protocol no. 2020/0056131 of 07/05/2020) for the interviews. According to Italian law, we were advised that approval for the anonymous survey was unnecessary. However, we were requested to adopt specific participant protection procedures in designing the survey (i.e., asking participants' interval-level information for age and workplace). The team provided written communication with the survey, including the aim of the study, confidentiality, the Italian Privacy Law statement, and the principal investigator's contact details. When the PTs returned the survey, informed consent was assumed. All the participants in the interview-based study provided written informed consent. Participants were only audio-recorded to ensure confidentiality during remote interviews. Interviews were conducted in settings guaranteeing privacy. Finally, the transcriptions were anonymized.

## 3 Results

### 3.1 Survey sample characteristics

The software used for the survey returned a total of 141 PTs. We cleaned the dataset from duplicate and invalid responses. PTs who were not treating cancer patients during the first wave of the lockdown were excluded as they were not eligible. The final dataset included 102 valid questionnaires (S1 dataset). Survey sample characteristics are summarized in Table 2. Most PTs were female (88.2%) and mainly aged between 41 and 60 (60.8%). Half of the PTs worked in the so-called "red zones" of northern Italy, where the virus spread peaked in March 2020. Nineteen (6%) PTs were subjected to compulsory quarantine during the weeks before the survey (March-May 2020). Cognitive and psychodynamic approaches were the most represented psychotherapeutic approaches used by PTs. Most (74.5%) had worked in psycho-oncology for at least six years, mainly treating adult cancer patients. Over half (57.8%) were employed in a hospital or a cancer research institute.

### 3.2 Interviewee sample characteristics

Thirty-nine PTs out of 102 consented to be interviewed by providing their contact details; 17 PTs chose not to participate in the study, citing reasons of being occupied with other commitments (n=11) or because researchers could not establish contact (n=6). One male and 21 females participated in the interview-based study. Most of them were 41–50 years old

**Table 2. Sociodemographic and professional characteristics of participants (survey respondents and interviewees).**

| Characteristics | | Survey respondents (N=102) n (%) | Interviewees (N=22) n (%) |
|---|---|---|---|
| **Gender** | Female | 90 (88.2) | 21 (95.4) |
| | Male | 12 (11.8) | 1 (4.6) |
| **Age** | <30 years | 3 (2.9) | 0 |
| | 31-40 | 37 (36.3) | 5 (22.7) |
| | 41-50 | 31 (30.4) | 12 (54.6) |
| | 51-60+ | 31 (30.4) | 5 (22.7) |
| **Macro-regions** | Northern Italy | 53 (52) | 13 (59) |
| | Central Italy | 23 (22.5) | 3 (13.6) |
| | Southern Italy and Islands | 26 (25.5) | 6 (27.4) |
| **Subjected to quarantine before survey's participation** | No | 82 (80.4) | 4 (18.2) |
| | Yes | 20 (19.6) | 18 (81.8) |
| **Psychotherapeutic approach** | Experiential | 16 (15.8) | 3 (13.6) |
| | Cognitive | 37 (36.6) | 4 (18.2) |
| | Systemic | 20 (19.8) | 9 (40.8) |
| | Psychodynamic | 29 (27.7) | 6 (27.4) |
| **Experience in psycho-oncology** | < 5 years | 18 (17.7) | 3 (13.6) |
| | 6-10 | 26 (25.5) | 3 (13.6) |
| | 11-20 | 28 (27.4) | 11 (50.2) |
| | 21-30 | 18 (17.7) | 2 (9) |
| | >30 years | 12 (11.7) | 3 (13.6) |
| **Type of patients** | Adults | 93 (91.2) | 22 (100) |
| | Children/adolescents | 9 (8.8) | 0 (0) |
| **Work context** | Hospital | 40 (39.2) | 10 (45.5) |
| | Cancer Research Institute | 19 (18.6) | 2 (9) |
| | Home care | 30 (29.5) | 8 (36.5) |
| | Hospice | 13 (12.7) | 2 (9) |

and worked in the northern regions of Italy. Almost all the interviewed PTs (n=18) were quarantined in March-May 2020. Half have been employed in the psycho-oncology field for at least 11 years. All were caring for adult cancer patients

### 3.3 Qualitative findings

Four themes were identified by analyzing the CIs and the semi-structured interviews: patient relationships, clinical practice management, emotional aspects, and organizational recognition. Several subthemes were also identified. The following sections describe emerging themes, subthemes, and illustrative quotations. Table 3 lists the themes and subthemes, with frequency across the PTs' CIs reported and intensity of labels. Figure 2 provides a thematic map to visualize the relationships between the different themes and subthemes.

**3.3.1 Patient relationships.** Most CIs highlighted the challenges cancer patients faced in maintaining family connections and communicating within the healthcare system, which PTs had to navigate and address in their practice. PTs noted the boundaries of patients' experiences, adapting their interventions to support patients in managing these difficulties, especially under the constraints of the pandemic. This required PTs to modify their therapeutic approaches to help patients cope with the strain on their relationships and the barriers to effective communication with healthcare providers.

**Table 3. Themes and sub-themes from qualitative analysis of CIs and semi-structured interviews, frequency of themes across CI's participants, and intensity of labels within themes.**

| Theme | Subthemes | Frequency and intensity of CI labels |
|---|---|---|
| Patients' relationship | • Impact of lockdown restrictions on family ties<br>• Impact of lockdown restrictions within the healthcare environment | 65/97 participants<br>34.67% of the labels |
| Clinical practice management | • The use of technologies for psychological interventions<br>• Timing and continuity of care<br>• Change in the number of psychological interventions | 56/97 participants<br>26.63% of the labels |
| Emotional aspects | • Emotions captured in patients<br>• Emotions captured in other professionals<br>• The emotional landscape of psychotherapists during the pandemic | 41/97 participants<br>14.24% of the labels |
| Organizational recognition | • Investments in psychological support and service coordination<br>• Issues about employment status | 27/97 participants<br>9.60% of the labels |

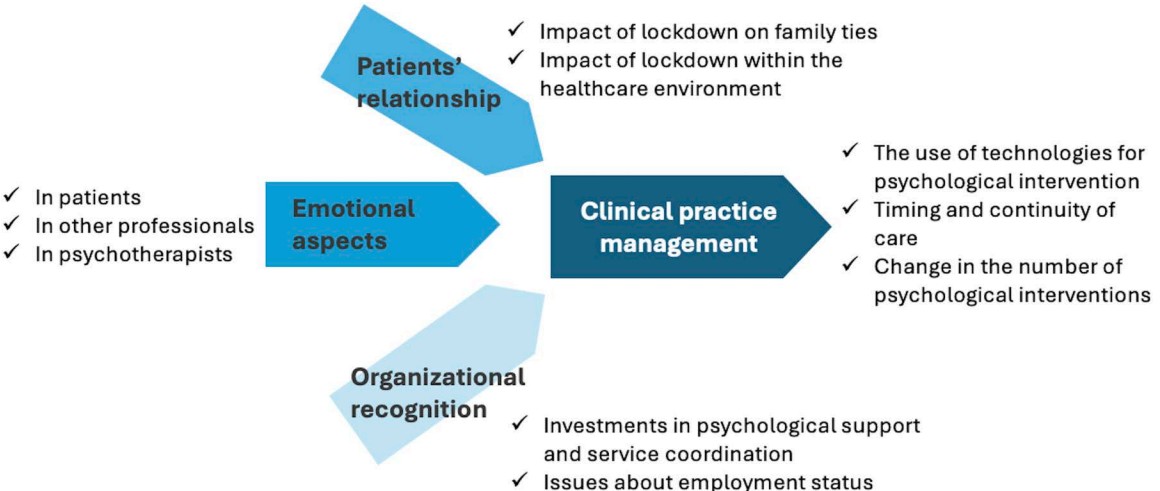

**Fig 2. Thematic map.**

**Role of restrictions on family ties** Most patients experienced profound loneliness due to social isolation, a challenge that PTs had to address in their practice. PTs noted that they often had to navigate the emotional impact of this isolation, adapting their care strategies and finding alternative ways to provide emotional support and maintain connection.

*"I greeted her. With a slight smile, she looked at me and said, 'Doctor, I'm happy to see you this morning. I really need to talk to you. […] This 'protective distancing' is forcing me to give up the embraces of my children who, for me, represent a therapy.'"* (participant 91_survey)

*"A patient with total laryngectomy who was unable to have family visits in the ward asked me if I could stay in his room to listen to a piece of Turandot with him."* (participant 89_survey)

The PTs also described new ways to approach their patients and maintain the therapeutic relationship through e-mails, phone calls, and remote sessions.

*"A young patient awaiting a mastectomy had a severe panic crisis when she learned that her partner […] was positive. […] She did not answer the phone or accept visits. So, I chose to try and initiate a written relationship with her.*

 

*After a few days, she replied with a long email. This became a regular appointment before using Skype."* (participant 73_survey)

Interviewed PTs reported stories of patients forced to live at home with other family members and burdened by intra-familial conflicts.

*"I had to cope with situations that were not easy […] For some people, staying at home was really hard because of previous family conflicts; many situations have worsened with the lockdown…"* (interviewee 15)

Only a few cases were reported among the CIs where the lockdown enabled family members to improve and reinforce their mutual relationships and made relatives more empathetic towards patients' experience of forced isolation due to immune suppression.

*"People already living with an oncological disease finally feel they can share their precarious experiences with others... sometimes reducing the perception of the severity of their situation."* (participant 32_survey)

The interviewed PTs highlighted the loneliness of hospitalized patients due to their distance from their families and the PTs' perceived role as mediators and guardians of family ties. They also discussed the extent to which the pandemic allowed patients to "decentralize" themselves from their disease, with beneficial effects on their psychological condition and family relationships.

*"I perceived myself as a sort of guardian, not of secrets but of intimate family stories, also with the task of reporting information to relatives who could not visit patients at the hospital. This aspect was emotionally draining but also beautiful."* (interviewee 17)

*"Patients think it is all about cancer. The arrival of a global pandemic has allowed patients to adopt a different point of view and to relativize things."* (interviewee 11)

**Role of restrictions within the healthcare environment**    Many CIs were negative, emphasizing the profound solitude patients faced in hospital wards, where "personal relationships, face to face, were eliminated," and care was sometimes described as "dehumanizing." PTs were deeply affected by witnessing these situations, particularly the distressing reality of patients dying alone in both hospitals and hospices. This intensified the emotional burden on PTs, leading some to advocate for more specialized home-care services, recognizing the need for alternative care models to address the isolation and improve end-of-life care.

*"Many hospice patients died without their family, with an intense sense of loneliness."* (participant 36_survey)

*"A young woman died alone in hospice, despite her desire to die at home. I was unable to intervene in time, discussing the case with my colleagues in the area"* (participant 59_survey)

Some PTs who could not enter the hospital due to face-to-face consultation suspension reported on the perceived indifference of the healthcare professionals towards cancer patients' psychological needs.

*"Once the face-to-face meetings with patients were suspended, they [healthcare professionals] showed very little consideration for the aspect of care related to psychological support in the ward."* (participant 86_survey)

Others felt like they were "intruding in the field" whenever they asked hospital staff for information about hospitalized patients they had in their charge.

The interviewed PTs underscored that they felt overburdened by choices concerning relatives' access to hospital or hospice and, in some cases, the unpleasant perception of not being able to work at their best.

*"We had to force ourselves to say 'No, you cannot stay here any longer' or 'No, your son cannot come into the hospital.' Thus, saying 'no' regarding something you intimately know is right"* (interviewee 17)

Positive CIs were few and mainly regarded the patients' increased motivation in dealing with their cancer treatments' clinical and psychological sequelae.

*"Considering the pandemic and lockdown as aspects that can allow patients to carry on their treatment and to deal with treatment side effects… This emerged from a psychotherapy session and reinforced the patient's willingness to focus on his therapy actively."* (participant 98_survey)

Some PTs also reported the warm atmosphere in certain hospital wards, where healthcare professionals promoted the patients' sense of being welcomed and regarded just "as in a family" despite their social isolation.

**3.3.2 Clinical practice management.** Clinical practice management represented the second most addressed theme within the CIs. This emerged from PTs' responses regarding three aspects: the challenge of handling new problems and using new tools for delivering psychological interventions, the need for preserving the timing and continuity of psychological care throughout the patient's pathway, and changes in the number of requests for psychological interventions during the pandemic.

**The use of technologies for psychological interventions** Challenges triggered by COVID-19 led the PTs to change their usual ways of conducting psychotherapy sessions. Many found remote tools to be a valuable solution, although the need for caution was highlighted. Sometimes, PTs reported that their initial discomfort with online interventions was soon replaced by the sense of "being together despite everything," as one PT emphasized.

Positive CIs included some surprising outcomes that PTs could not otherwise have experienced in the traditional setting of in-person clinical sessions and significant disclosures from patients, allowing them to deal with issues not previously accessible within their psychotherapy treatment.

*"Where the therapeutic relationships were strong, I have found significant patient disclosure. We were able to touch upon profound issues. With the 'classic' setting, this had not been possible."* (participant 47_survey)

The disclosures observed by PTs from patients were not simply expressions of existential dread or fear of death. Instead, they reflected deeper, previously unexplored emotional and psychological insights shaped by the context and the use of remote tools. PTs highlighted that the virtual setting and the unique challenges of the pandemic enabled conversations about highly personal issues that patients had not previously addressed in face-to-face therapy sessions.

Additionally, these new modalities seemed to increase the chance of providing individuals with psychological interventions in specific situations, as in the case of a PT caring for a bereaved relative who had moved to another region.

*"Through the telephone interviews, I could remotely follow the caregiver of a bereavement patient, who has moved to another region. In a 'normal' situation, I couldn't provide this kind of support."* (participant 19_survey)

Conversely, some PTs reported that clinical sessions via the web and telephone were sometimes difficult and tiring for patients, especially those in frail physical condition.

*"The lockdown and the possibility of temporarily continuing the treatment via telematics require us to proceed cautiously. There are limitations to managing the impact of what can emerge from in-depth work."* (participant 25_survey)

In some CIs, collaboration with the ward staff to make online psychotherapy sessions feasible for hospitalized patients emerged as paramount.

The interviewed PTs widely underlined the challenge of using remote tools in terms of users and their reactions and prejudices, methodological aspects to pay attention to when delivering clinical sessions, usable and unusable psychotherapeutic techniques, and privacy issues.

*"Thinking about those months, I will carry with me the discomfort I felt during hours spent at the computer, on the phone, or in front of a screen, without the possibility of a different relationship that I was used to."* (interviewee 15)

*"My difficulty was not related to technology but to the relationship. I practice body psychotherapy, so not having physical contact with patients was strongly limiting for me, at least at the beginning."* (interviewee 21)

*"At the beginning, I was a little bit worried, then I understood that I could give myself more time, especially with new patients."* (interviewee 13)

**Timing and continuity of care**  Guaranteeing both the continuity of care and the timing of psychological interventions represented pivotal issues. Positive CIs highlighted patients' reactions concerning these topics, such as their relief about not being abandoned by their PTs and the opportunity to use remote tools to continue their psychotherapy treatment during the lockdown.

*"We proposed to our patients that they continue our work by phone or video call. The perception of a meaningful and engaging experience soon replaced the initial uncertainty. […] Almost all my patients showed themselves extraordinarily able to cope with this new approach. A recurring statement was: "If it had happened to me before cancer, I would have panicked!"* (participant 69_survey)

*"Parents [of cancer patients] did not expect to have the same services that they had always been used to before the lockdown. New patients have been taken on, clinical sessions have been regularly carried out, and play-based activities with ill children and adolescents."* (participant 92_survey)

One PT reported that the timing allowed a patient to process a recently occurring diagnosis more quickly. Another PT disclosed a situation where he was exceptionally authorized to deliver a psychotherapy session at a patient's home; since the patient could not communicate remotely and his clinical condition had rapidly deteriorated, exploring where he wished to die was essential.

*"A home visit to a patient who has been diagnosed with both cancer and ALS. Communicating online was impossible for him […]. As his condition worsened, it was necessary to find out his preference for the place of death […]. The patient's wishes were collected and respected. […] We could say goodbye to each other with dignity."* (participant 102_survey)

Regarding negative CIs, PTs widely highlighted the unpleasant consequences of the forced interruption of their routine psychotherapy treatment, especially in patients with advanced illnesses. In some cases, patients refused online meetings and preferred to wait for face-to-face therapy sessions. The choice of not addressing complex issues with new patients taken on during the lockdown was argued by some PTs who, in selected cases, preferred to build just a basis for a therapeutic relationship while waiting for the feasibility of in-person sessions.

*"We deliver both individual and group psychotherapy sessions. Ongoing groups were abruptly terminated. People felt abandoned and left in limbo. After one month, facilitators tried contacting participants to alleviate their feeling of being abandoned. […] Now we are resuming group sessions, respecting the distancing rules."* (participant 69_survey)

PTs widely reported the extent to which interprofessional collaboration, searching for scientific information, and input from the literature were helpful in building strategies for guaranteeing continuity of care.

*"There was a constant connection with friends, colleagues, physicians, nurses, teachers… to discuss the choices to be made. Free time was occupied because it was filled with all these things."* (interviewee 20)

**Changes in the number of requests for psychological interventions**    Most PTs reported a general increase in requests for psychological interventions. Some highlighted that, during the lockdown, psychological support was also requested by those previously reluctant to receive it.

In turn, what happened was that patients who had previously achieved an excellent adjustment to cancer illness asked the PTs to resume their psychotherapy sessions to address their pandemic-related distress.

*"Some patients who had adapted to the oncological disease felt the need to resume psychotherapy sessions due to the presence of elevated symptoms of anxiety and depression."* (participant 84_survey)

In the interviewed PTs' responses, this phenomenon was linked to patients' increased worries about delays in scheduling their cancer treatments and follow-up examinations.

The interviewed PTs also highlighted an increased request for psychological support from relatives, particularly in the bereavement phase, and from healthcare professionals burdened by work overload.

Only in a smaller number of cases was a decrease in requests for psychological support reported.

**3.3.3  Emotional aspects.**  Almost half of the PTs reported dealing with feelings toward the pandemic. They referred to this topic, focusing on emotions captured in patients, families, and other professionals. In addition, some revealed their inner emotional world facing the pandemic as human beings and PTs.

**Emotions captured in patients and families**    Overall, the PTs described patients' emotional conditions in terms of fear and anxiety.

*"From a look in the waiting room, when patients were waiting for chemotherapy, I saw the eyes of patients so listless and afraid. They no longer communicated with each other; a veil of frost enveloped them. What was threatening their existence was not only cancer but another silent enemy..."* (participant 91_survey)

The PTs described a range of negative emotions among patients related to the healthcare environment, where hospitalized patients perceived themselves as abandoned, unseen, or neglected by healthcare professionals.

*"A very critically ill young patient, in treatment with chemotherapy, was "forced" to go to Milan to do a CT scan and other tests... I wonder if it was not possible to avoid this stress. She is almost dying now and not due to COVID-19."* (participant 71_survey)

Regarding positive CIs, as mentioned above, PTs referred to social isolation as a common condition that could bring cancer patients closer to their relatives. Some found their patients relieved by this in terms of not being considered "special" anymore, whereas "being special" referred to their experience of prolonged social isolation due to immune suppression.

The interviewed PTs highlighted emotions experienced by relatives, such as fear that the pandemic could cause delays in treatment or follow-up examinations.

**Emotions captured in other professionals**    PTs frequently focused their responses on emotions their colleagues and other healthcare professionals perceived.

*"Helping a nurse who had had a suicide case among her patients. Frustration and guilt dominated her days."* (participant 28_survey)

*"The situation involved the entry of patients from resuscitation. These patients had characteristics of clinical fragility that put the team's ethical decisions in a crisis. Often fractures arose between physicians and operators, and they failed to share constructively."* (participant 92_survey)

PTs also highlighted healthcare professionals' inner resources and competencies, specifically their ability to cope with impactful situations and ethical dilemmas arising from clinical decision-making.

Inter-professional collaboration, sharing thoughts and feelings, and supervision sessions emerged from the CIs as valuable means for dealing with negative emotions experienced by healthcare staff during an emergency.

During the interviews, some initiatives aimed at supporting PT colleagues and other professionals were reported; these stemmed from ongoing assessments of professionals' reactions and psychological needs throughout the lockdown.

*"Ward professionals had been allowed to participate in EMDR group sessions. Single professionals who needed individual treatment received individual sessions. Sometimes professionals themselves expressed their needs by email."* (interviewee 13)

*"I am a curious person, so I had read that the NCCN published extracts on how oncology professionals could cope with distress during the lockdown. I translated these and added some images. Then, I sent the document via WhatsApp to physicians, nurses, and nursing assistants. [...] I have organized groups of 5-6 professionals who met twice or thrice weekly for an hour. People talked, played down the situation, and expressed how they had to face this dramatic situation, their resources, and their fear."* (interviewee 16)

**The emotional landscape of psychotherapists during the pandemic**    Many PTs shared feelings of fear, anxiety, and vulnerability, recognizing that they were not immune to the same concerns as their patients regarding the risk of contracting the virus. This shared vulnerability sometimes heightened their emotional response to their professional duties. For some PTs, the suspension of psychotherapy sessions led to feelings of helplessness and frustration. They expressed discomfort with missed opportunities to provide care, particularly in critical moments, such as when patients deteriorated or died without their psychological support.

*"And it was harrowing not to have been able to say goodbye to a patient who died a few days ago after more than three years of psychological support. Suddenly, she was hospitalized. And I, working from home, could not go to the ward to support and talk to her."* (participant 46_survey)

*"I have felt that I "missed" an important appointment on many occasions. Patients go to the hospital, but we are not there."* (participant 52_survey)

The transition to online therapy sessions also presented emotional and practical challenges. PTs felt disconnected, describing the remote work environment as an undefined, isolating space where they struggled to maintain the emotional engagement they typically cultivated in face-to-face sessions. They felt a sense of immobility as if they could not fully perform their role due to the absence of the physical presence, therapeutic setting, and embodied interaction they relied on in their practice. Additionally, PTs spoke about the strategies they developed to manage their emotional well-being while continuing to provide care. For some, maintaining a sense of order amidst the uncertainty of the pandemic required significant emotional self-regulation.

*"There were regional, provincial, and corporate directives to be respected. The goal was to try to make order out of the chaos, and I had to work on myself to keep my balance."* (interviewee 21)

                                                                                 

Through these accounts, PTs were not only attending to their patients' needs but also grappling with their emotional struggles, using self-care strategies to navigate the difficulties posed by the pandemic.

### 3.3.4 Organizational recognition.

**Investments in psychological support and service coordination**    PTs who delivered psychological support to healthcare teams during the outbreak reported on the need to establish such programs as a service to be permanently introduced in the healthcare practice, and not only during health emergencies. In some cases, they reported how decision-makers within their organizations undervalued the potential benefits of their work, an issue strongly stressed by the interviewed PTs, and the need to respect the culture of psycho-oncology and to guarantee psychological interventions for cancer patients and their families during and beyond the emergency.

*"For the management of similar emergencies, I believe it is important to consider psycho-oncological support as a fundamental activity in patient care, part of a multidisciplinary care."* (participant 13_survey)

*"Psychologists are always seen as superfluous, even though they must be available in certain situations. I don't always have a room where I can receive my patients. A volunteer's things often occupy my office…"* (interviewee 1)

Some PTs complained about a lack of integration among services. Developing a network of services both locally and at a national level was also advocated to optimize psychological interventions for cancer patients during emergencies.

*"National coordination is needed to promote psychologists' profession in public and private contexts. A sense of psychotherapeutic culture and the propensity of people to ask for psychological help are still lacking."* (participant 15_survey)

**Issues about employment status**    Although marginally, some surveyed and interviewed PTs reported that the lockdown impacted their employment conditions, highlighting their distress linked to short or expiring contracts and the event of unpaid remote work.

*"I would like the possibility of a permanent job, and not with an annual contract. Especially after 40 years of service."* (participant 11_survey)

*"I work as an employee manager. I can work overtime and accumulate the extra hours I work. Nevertheless, it is very difficult to recover them."* (interviewee 12)

## 4 Discussion

### 4.1 Summary of findings

To the best of our knowledge, this study, which was performed on a large sample of PTs from several Italian regions, is the first to focus on the practice of psychotherapy addressed to cancer patients during the first phase of the pandemic. We employed a qualitative method to gather information from the PTs' "real world" and through their voices. As reported elsewhere [26–28], this allowed the researchers to obtain data concerning the PTs' direct experiences rather than theoretical reflections or opinions.

Qualitative analysis performed on both the CIs and the semi-structured interviews led us to group the PTs' experiences and practices into four theme-based categories, each including specific subthemes: patient relationships (the impact of restrictions on family ties, the impact of restrictions within the healthcare environment), clinical practice management (new problems and tools for psychological interventions, timing and continuity of care, changes in the number of requests for psychological interventions), emotional aspects (emotions captured in patients, emotions captured in other professionals,

the emotional landscape of psychotherapists during the pandemic), and organizational elements (investments in psychological support and service coordination, issues related to employment status). Specific themes and subthemes from our qualitative analysis are reminiscent of those identified in previous research on psycho-oncology professionals during the pandemic, even though these prior studies employed diverse methodological approaches. In the study performed in the UK by Archer and colleagues [14], participants reported concerns about the disruption of psychological care delivered to patients, a reduction in patients referred to psycho-oncology services, and worries about the ability to provide the same standard of care through remote methods. Nevertheless, the drive to implement improvements in working practice also emerged through reducing process barriers and fostering inter-professional collaboration. Costantini and colleagues [5], in their study aimed at assessing Italian psycho-oncologists' well-being, found that moderate and severe peritraumatic distress levels were associated with living alone and the perception of being avoided by family or friends because of work. The subject of challenges linked to the use of remote consultations was the focus of three studies. In the phenomenological research conducted by Morgan and their team in the UK [13], participants detailed how the practice of remote consultations evolved, transitioning from an initial crisis state to a new normal. In a Dutch study led by van der Lee and collaborators [2], video consultations emerged as a viable alternative when in-person meetings became overly burdensome for patients and as a promising tool with potential utility beyond the pandemic, particularly for patients who are too frail to travel or require motivation for therapeutic exercises in some psychotherapies. Similar findings were documented in a study conducted in the UK by Millar and their colleagues [12], where participants also highlighted the high rate of patients receiving psychological interventions via remote consultations and the relief expressed by patients who felt validated in avoiding healthcare settings associated with the trauma of illness.

From our findings, three main issues characterizing the delivery of psychological interventions to Italian cancer patients during the COVID-19 emergency can be outlined, which we will discuss in the following sections.

### 4.2  The challenge of delivering psychotherapy through information and communication technologies

The first issue concerns the role of information and communication technologies (ICTs) in the organizational changes made within the health system during the pandemic. These changes, which also concerned oncology and palliative care settings, raised significant questions about the continuity of psychological care for cancer patients. On the one hand, the suspension of in-person psychological interventions challenged PTs' ability to be a source of professional support for their patients [13,27]. On the other hand, such service restrictions changed how patients experienced and accessed psychological services. As reported elsewhere, the surveyed PTs initially felt displeased about suspending their clinical activity [12]. Nonetheless, a new way to approach patients and maintain therapeutic relationships became possible thanks to the prompt use of ICTs, a measure strongly suggested in Italy soon after the lockdown and implemented in many other countries [27].

Although online ICT was considered an opportunity [12,29] and a means to reconnect with patients [2], the shift from face-to-face to online meetings caught many PTs off guard. As highlighted elsewhere, they sometimes felt they needed to be more skilled and comfortable using digital devices [14,27]. They questioned their ability to deliver the same standard of care effectively to their patients. Recent studies on cancer patients and healthcare professionals have widely discussed this topic, emphasizing the strengths and limitations of ICT employment in clinical consultations during the COVID-19 pandemic [26,30–32]. According to Clerici and colleagues, ICT can be helpful within an established doctor-patient relationship but is very difficult for new patients [26]. Millar and colleagues outlined that patients and therapists adapted well to these changes [12].

Since the first wave of the pandemic, the increasing employment of ICTs represented (and still means) an innovative way of delivering psychotherapy within both institutional and private contexts [10]. Promptly, PTs gleaned from the pandemic that fostering a sense of "closeness" and "presence" was achievable even through a screen or a phone. Nevertheless, it is essential to acknowledge that these tools are not neutral. During communicative exchanges, they significantly

restrict the perception of time and space, physical postures, and interpersonal distances. Additionally, they limit body language and eye contact while omitting other non-verbal socio-emotional cues like scents, physical contact, and sensory input from the environment. Individuals may compensate for these limitations by filling the information gap with their imagination, which is inherently tied to their past experiences and expectations. However, all the factors mentioned above inevitably impact the efforts of PTs in establishing and maintaining authentic relationships based on empathy and compassion that form the cornerstone of every psychological intervention.

As highlighted elsewhere [33], some measures can be implemented to mitigate the inherent constraints of video consultations. For example, selecting a noise-free environment with a stable internet connection and positioning the webcam at an appropriate distance from the patient's image on the screen to enhance eye contact and ensure proper framing of one's face. During these communicative interactions, it is essential to be attentive to fluctuations in one's vocal tone and the patient's, facilitating the establishment and maintenance of emotional connection and resonance.

Thus, future research on this topic should address the following issues, i.e., how remote consultations could impact therapeutic goals, how the quality of the therapeutic alliance can be effectively ensured, and the extent to which strategies and tools from different psychotherapeutic approaches may continue to be used or need to be adjusted to maintain their efficacy in the context of a digital environment [2,33]. In addition, the acceptability and feasibility of online psychological interventions for cancer patients is still a matter for further investigation despite evidence suggesting the positive effects of such an approach on these patients' perceived support, knowledge, and information competence [34]. Another point concerns the need for specific training on the potential advantages of ICTs and how to conduct online consultations, which are starting to be implemented in webinars.

### 4.3  The "bridging" role of psychotherapists between different users and healthcare settings

A second issue that emerged from our findings is what can be defined as the "bridging" role of PTs. Our PTs described themselves as collectors of multiple support requests. The needs of cancer patients related to the pandemic increased their pre-existing psychological burden, thereby complexifying PTs' work for an indefinite period, as has already been suggested [2,8,10,33]. In some cases, hospital-based PTs were the only people with whom patients could relate when Italian oncological departments were reorganizing their service provision. Many were involved in national helpline activities to support bereaved relatives or were asked to help other healthcare professionals with their experiences related to work overload and emotional burdens. In this way, by detecting and responding to both users' and providers' emotional issues and psychological needs and sustaining the interpersonal relationships limited by lockdown restrictions, the PTs acted as a bridge between all the actors involved in the care process and within different healthcare settings. Other studies also highlight this role in describing how COVID-19 impacted psycho-oncology service provision [2,14,26]. In the picture of PTs' emotional condition and inner resources during the pandemic provided within the cross-sectional web-based study performed in Italy in 2020, the surveyed PTs showed greater resilience and ability to maintain a positive vision of the future than other healthcare professionals. Indeed, PTs are trained to help others, deal with cancer-related limitations, and review their values considering stressful changes. The authors propose that psycho-oncology associations should play a role in implementing policies designed to foster a sense of social connectedness. This can be achieved by offering interactive guidance and a scientific engagement system [5].

### 4.4  The poor recognition of psycho-oncology

The third issue concerned the lack of professional recognition of PTs' role in their increased workload during the pandemic and the need to ensure psychological interventions are integrated with other healthcare services and coordinated at a national level. In our study, many PTs complained about a lack of recognition of the value of their work, despite the perceived increase in their clinical activities with cancer patients, their relatives, and healthcare staff, and the positive feedback from users concerning the benefits of the psychological interventions they were provided with during this time.

Given the fundamental role of cancer-related psychological support not only during but also beyond health emergencies and outbreaks [7,8,28,35,36], this pandemic may represent an impetus for the effective integration of psycho-oncology into organizations, as widely recommended by international and national guidelines (please see www.siponazionale.it) [16,36,37]. The results of a survey recently performed by SIPO highlight how psycho-oncology services are not uniformly distributed across Italy; furthermore, these services are heterogeneous both in terms of the activities performed and the number of PTs involved (please see www.siponazionale.it). As in many other countries worldwide, even in Italy, the core curriculum of the psycho-oncologist has not yet been clearly defined, nor is the training path needed for the certification of the psycho-oncologist profession [38].

Nonetheless, in the last two years, some necessary actions have been taken to promote and enhance psycho-oncology in our country. In July 2022, the Regional Council of Lazio approved the first regional law on psycho-oncology, which establishes the criteria to promote psycho-oncological care services within the regional cancer network [39]; presently, the process for the proposal or approval of a law on psycho-oncology has been started or is underway in other four Italian regions. In October 2022, a national bill for establishing and strengthening psycho-oncology services was presented and is currently being examined by the Chamber of Deputies Health Commission [40]. The bill represents a significant turning point for the development of psycho-oncology in our country. It recognizes the importance of psychological intervention for cancer patients, their families, and health professionals and systematizes such intervention as an integral part of the care pathway. According to the bill, the psycho-oncologist must be included within all hospital oncology and once-hematology units, participate in the multidisciplinary healthcare teams, and be involved in the Diagnostic and Therapeutic Care Pathways. Besides, the training required to carry out the psycho-oncologist profession is initially defined as the first step for developing a core curriculum and establishing a specific health profile for the Ministry of Health. We believe that the approval and implementation of this bill, as the result of a long process toward raising awareness of the role of psycho-oncology in the bio-psycho-social approach to cancer care, could give an essential boost to the development of this discipline and provide an appropriate response to the unmet psychological support needs of many cancer patients and their families.

### 4.5. Strengths and limitations

Findings from this study should be interpreted considering its strengths and limitations. In this mixed-methods research, we employed a convenience sampling approach, which introduces selection bias due to participation's voluntary and self-selected nature [41]. PTs who responded to the survey and interviews did so voluntarily, meaning our sample may not fully represent all PTs working with cancer patients in Italy. Consequently, our findings reflect the experiences of those who chose to participate, potentially omitting perspectives from those less inclined to engage in research, particularly those underrepresented in some regions of psycho-oncology.

Most of the participating PTs worked with adult cancer patients, limiting the generalizability of the findings to those working with pediatric oncology populations. This lack of representation from psychotherapists specializing in pediatric patients is a significant limitation, as pediatric oncology's challenges and therapeutic approaches may differ considerably from those in adult settings. Furthermore, the regional concentration of PTs, particularly those from the "red zones" heavily impacted by the initial wave of the pandemic, may have skewed the data toward experiences in regions the most affected by COVID-19. This regional bias limits the ability to generalize findings to PTs in areas with differing levels of pandemic impact.

Despite these limitations, our study gathered valuable insights from a nationwide pool of PTs representing diverse psychotherapy approaches and substantial experience in psycho-oncology. The data collected through the survey were further enriched by qualitative individual interviews, providing a comprehensive understanding of PTs' experiences and practices during the pandemic. Although the sample may not fully represent all PTs involved in the survey, the findings offer important lessons and reflections on the challenges faced and strategies employed by PTs during public health crises. These insights can contribute to better preparedness and support for PTs in future emergencies.

## 5 Conclusions

This study offers important insights into the challenges faced by PTs during the COVID-19 pandemic and highlights critical areas for improving clinical practice and informing health policy. Based on our findings, several actionable recommendations can be proposed to enhance psycho-oncology services and their integration into healthcare systems.

The rapid shift to remote psychotherapy via ICTs during the pandemic revealed opportunities and limitations. Health systems should prioritize developing and implementing targeted training programs for PTs to enhance their digital competencies. This includes technical proficiency and strategies for maintaining therapeutic presence and empathy in virtual environments. Additionally, standardized protocols for ICT use in psycho-oncology should be developed, ensuring that remote consultations meet clinical quality standards and patients' emotional needs. To ensure continuity of care during future health crises, psycho-oncology services should incorporate ICTs, with adequate resources and infrastructure, as a core part of their service provision. Policymakers should support adopting hybrid care models that combine in-person and digital care, allowing flexibility in service delivery based on patients' needs and pandemic-related restrictions. This could include integrating telehealth platforms within national health services and private healthcare institutions.

Given the emotional strain on PTs during the pandemic, ensuring they have access to support systems is crucial. Institutions should establish formal mental health and peer-support networks for PTs, providing regular supervision and emotional debriefing sessions. This will help sustain PTs' well-being and resilience, which is critical for maintaining the quality of psychological care for cancer patients during crises. In addition, the crucial role played by PTs in supporting patients, families, and healthcare professionals during the pandemic underscores the need for better integration of psycho-oncology within national and international health policies. Governments and healthcare institutions should effectively recognize psycho-oncology as a critical component of cancer care. This can be achieved by including psycho-oncology services in national cancer treatment guidelines and emergency preparedness plans, ensuring sufficient funding and resource allocation for these services.

Many PTs in our study expressed concern over the lack of institutional recognition for their work during the pandemic. Health policymakers should advocate for formally recognizing psycho-oncology within healthcare systems, ensuring PTs are adequately compensated and supported. This includes allocating specific funding for psycho-oncology services, particularly during public health emergencies, and establishing career development pathways to attract and retain skilled PTs.

By implementing these recommendations, healthcare systems can better prepare for future emergencies, ensuring that psycho-oncology services remain robust, flexible, and capable of addressing the complex psychological needs of cancer patients, families, and healthcare providers. These measures will ultimately enhance the quality of psycho-oncology care and promote the mental well-being of all those involved in cancer treatment.

## Supporting information

**S1 dataset. Final dataset of valid questionnaires.**
(XLSX)

## Acknowledgments

The authors thank all the participants who gave their time to share critical events impacting Italian psycho-oncology. We also thank the Italian Society of Psycho-Oncology (SIPO), the Italian Society of Palliative Care (SICP), and the Zero K Association for supporting the survey dissemination.

## Author contributions

**Conceptualization:** Luca Ghirotto, Ludovica De Panfilis, Silvia Di Leo.

**Data curation:** Alessandra Miraglia Raineri, Francesco De Vincenzo, Matías Eduardo Díaz Crescitelli.

**Formal analysis:** Marta Perin, Francesco De Vincenzo, Matías Eduardo Díaz Crescitelli, Elisa Rabitti.

**Investigation:** Ludovica De Panfilis, Alessandra Miraglia Raineri, Silvia Di Leo.

**Methodology:** Luca Ghirotto, Silvia Di Leo.

**Project administration:** Marta Perin, Francesco De Vincenzo, Matías Eduardo Díaz Crescitelli.

**Supervision:** Luca Ghirotto.

**Validation:** Ludovica De Panfilis, Silvia Di Leo.

**Visualization:** Marta Perin, Francesco De Vincenzo, Elisa Rabitti.

**Writing – original draft:** Luca Ghirotto, Ludovica De Panfilis, Matías Eduardo Díaz Crescitelli, Silvia Di Leo.

**Writing – review & editing:** Luca Ghirotto, Marta Perin, Alessandra Miraglia Raineri, Francesco De Vincenzo, Elisa Rabitti.

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
