## [Decision Letter · Decision Letter 0]

14 Oct 2024

PONE-D-24-04536Psycho-oncology practice for cancer patients during the pandemic lockdown: a qualitative mixed-method study with Italian psychotherapistsPLOS ONE

Dear Dr. PERIN,

Thank you for submitting your manuscript to PLOS ONE. After careful consideration, we feel that it has merit but does not fully meet PLOS ONE’s publication criteria as it currently stands. Therefore, we invite you to submit a revised version of the manuscript that addresses the points raised during the review process.

We look forward to receiving your revised manuscript.

Kind regards,

Jan Christopher Cwik, Ph.D.

Academic Editor

PLOS ONE

Journal Requirements:

2. In the online submission form, you indicated that [All relevant data are within the manuscript and its Supporting Information files. The data supporting this study's findings are available from the corresponding author upon reasonable request.].

Reviewers' comments:

Reviewer's Responses to Questions

**Comments to the Author**

1. Is the manuscript technically sound, and do the data support the conclusions?

Reviewer #1: Yes

Reviewer #2: Partly

2. Has the statistical analysis been performed appropriately and rigorously? 

Reviewer #1: N/A

Reviewer #2: Yes

3. Have the authors made all data underlying the findings in their manuscript fully available?

Reviewer #1: Yes

Reviewer #2: Yes

4. Is the manuscript presented in an intelligible fashion and written in standard English?

Reviewer #1: Yes

Reviewer #2: Yes

5. Review Comments to the Author

Reviewer #1: [Response below is found in attachment as well]

Thank you for giving me the opportunity to review this manuscript about this pivoting time in psycho-oncology history. The article was well-written and multi-layered. There were many relevant aspects that had important potential in terms of exploring the role and importance of psychotherapy within hospital settings.

Within the abstract, I suggest you change “the final dataset included 102 valid questionnaires” to “the final dataset included 102 valid responses”.

The introduction is clear although may be unnecessarily lengthy. I suggest that lines 65 to 81 be shortened to a sentence or two as I don’t think it is relevant to your research question. In addition, on lines 101-104 you mention that your article’s overarching goal is to promote psycho-oncology program. It is unclear to me in what ways this article will achieve this. Although I agree that your article advocates for more recognition of psycho-oncology professionals within hospital settings, have you thought about the ways in which you will disseminate your findings to stakeholders and policy makers within these hospital settings? Implementation science is the next big thing; how will your research contribute to the implementation of psycho-oncology care? Finally, should you have space, it may be important to further introduce/explain/breakdown what CIT is.

The methods are generally clear and straightforward. However, as mentioned above, it may be relevant to provide further explanations as to your used of CIT. I suggest you also provide more information on your use of triangulation. It may be worth, for reproducibility, to provide more details on how this “framework” was used/built. Finally, it is unclear to me why you asked your participants about previous involvement in emergency psychology. How does that impact their answers to your questions? In what ways is this important to keep in mind? Finally, it may be important to clarify that this is an inpatient setting, right from the beginning, as the situation may be vastly different in outpatient settings.

The results are well outlined, and themes appear clear and distinct. I suggest making theme 1 more explicit in the ways it relates to PTs’ practice and how they adapted. It appears to me that it is more strongly related to patients’ experience rather than PTs’ and therefore does not add much to your research question. In addition, the disclosure from patients could be further explained. How is this different than existential dread when facing potential death?

Finally, the discussion really ties it all together. I appreciated the clearer distinction between your two bigger themes: innovation and lack of recognition. I believe both have great potential but are too tangled in your results. I suggest taking this distinction which you have mastered in the discussion and applying it to the rest of the article. In addition, most themes seem to be rooted in systemic issues and situational factors. Can you further discuss the recognition of psychotherapy in hospitals using a public health or systemic lens? What do those results mean, what changes are needed? You may not need to spend as much time on ICTs.

Overall, this article addresses some important issues within psycho-oncology and sheds some light on its in hospital settings. Thank you for your submission.

Reviewer #2: Thank you for the opportunity to review your manuscript. The topic you explore is highly relevant, especially given the challenges introduced by the COVID-19 pandemic in psychological support practices within oncology. Below, I provide a detailed review of your manuscript along with several suggestions to further strengthen it.

General Comments

The manuscript is well-written and addresses a timely and important topic. The use of a mixed-methods approach is appropriate for answering your research question. However, there are a few methodological and conceptual aspects that could be expanded to improve the transparency and clarity of the work.

1. Title and Abstract

-The title is clear and descriptive, but I recommend specifying the geographic context (Italy) to make the focus on Italian psychotherapists more explicit for an international audience.

-The abstract is clear, but the methodology section could be expanded, particularly to explain the "QUAL→qual mixed-method" approach for readers unfamiliar with this terminology.

2. Introduction

-The introduction provides a comprehensive background on psycho-oncology and the implications of the pandemic. The reference to the history of psycho-oncology in Italy is well done.

-Provide a clearer statement of the research gap early in the introduction.

-Better explain the rationale for focusing on the Italian setting by relating it to international contexts, which could make the study’s findings more broadly applicable.

3. Methods

-The mixed-methods approach (QUAL→qual) is appropriate, but more details are needed about how the qualitative data were triangulated and what criteria were used for selecting participants for the interviews.

-Convenience sampling is acknowledged as a limitation, but the manuscript would benefit from a more in-depth discussion of how this might affect the generalizability of the findings.

-Provide more information about the coding process, such as how discrepancies between coders were resolved.

Clarify whether any measures were taken to ensure that the interview sample was representative of the broader survey population.

4. Results

-The four main themes (patient relationships, clinical practice management, emotional aspects, and organizational aspects) are well supported by participant quotes, but the results section would benefit from clearer separation between survey and interview data. Reorganize the results section to explicitly separate the findings from the survey and interview phases before integrating them in the discussion.

-Some subthemes, such as “the inner world of the psychotherapists,” are more abstract and would benefit from further elaboration or concrete examples.

5. Discussion

-The discussion reflects thoughtfully on the challenges psychotherapists faced during the pandemic and effectively ties the findings to existing literature.

-The discussion could benefit from a more critical evaluation of the study’s limitations, especially concerning the convenience sampling and the lack of representation from psychotherapists working with pediatric patients. Expand the discussion with a more detailed analysis of how these limitations might have influenced the results and their generalizability.

-The points on the importance of ICT tools and staff support programs are well made, but the discussion could offer more concrete recommendations for policy and practice, particularly concerning how psycho-oncology services should prepare for future crises.

6. Conclusion

-The conclusion effectively summarizes the key contributions of the study, particularly the value of ICT tools and the need for ongoing support for psycho-oncology services. Provide more actionable recommendations on how your findings can inform clinical practice or health policy. This would help bridge the gap between research and practice.

7. Ethical Considerations

-Ethical approval is well described, and the consent process is clear. However, it would be helpful to briefly address any challenges related to ensuring confidentiality during remote interviews, given the sensitive nature of the topic.

8. Figures and Tables

-The figures and tables are well-organized and informative. Table 2 is particularly helpful in contextualizing the sample. Consider adding more detail to Figure 1 (Mixed-methods design process) to clarify how the different phases of the study were integrated in the analysis.

-Include thematic maps to better visualize the relationships between the different themes and subthemes.

6. PLOS authors have the option to publish the peer review history of their article (what does this mean? ). If published, this will include your full peer review and any attached files.

**Do you want your identity to be public for this peer review?** For information about this choice, including consent withdrawal, please see our Privacy Policy .

Reviewer #1: No

Reviewer #2: No

---

## [Author Response · Author response to Decision Letter 1]

27 Nov 2024

Dear Editor,

Please find enclosed the article entitled: “Psycho-oncology practice for cancer patients during the pandemic lockdown in Italy: a qualitative mixed-method study with psychotherapists” that we are re-submitting for publication in PlosOne.

In this cover letter, we explain the revisions made. Changes are tracked within the manuscript. Together with the co-authors, we have discussed the comments and revised the text accordingly.

We thank the Editor and the reviewers for their time, careful review, and constructive feedback.

We hope we have met the expectations, and we are willing to provide any further clarification as needed.

Our reponse to Editor and Reviewers' comments have also been provided as a separate file labeled 'Response to Reviewer' whithin the Attach Files.

Best regards,

Marta Perin and colleagues

REVIEWER 1

Thank you for giving me the opportunity to review this manuscript about this pivoting time in psycho-oncology history. The article was well-written and multi-layered. There were many relevant aspects that had important potential in terms of exploring the role and importance of psychotherapy within hospital settings.

RESPONSE: Thank you for your time and effort in reviewing our manuscript. We would like to clarify a possible misunderstanding concerning the research settings. We involved psychotherapists who served in psycho-oncology practice in different setting including hospitals, but not exclusively

Within the abstract, I suggest you change “the final dataset included 102 valid questionnaires” to “the final dataset included 102 valid responses”.

RESPONSE: Thank you. We amended it.

The introduction is clear although may be unnecessarily lengthy. I suggest that lines 65 to 81 be shortened to a sentence or two as I don’t think it is relevant to your research question.

RESPONSE: We rephrased and shortened the section accordingly.

In addition, on lines 101-104 you mention that your article’s overarching goal is to promote psycho-oncology program. It is unclear to me in what ways this article will achieve this. Although I agree that your article advocates for more recognition of psycho-oncology professionals within hospital settings, have you thought about the ways in which you will disseminate your findings to stakeholders and policy makers within these hospital settings? Implementation science is the next big thing; how will your research contribute to the implementation of psycho-oncology care?

RESPONSE: Thank you for this insightful comment. Indeed, the aim of this study is exploring the experiences and practices implemented by PTs who provided care to cancer patients during the initial wave of the COVID-19 pandemic in Italy; we agree that the overarching goal we mentioned is not actually within the aims of our study. Rather, we believe that our findings could indirectly contribute to the promotion of psycho-oncology programs by providing both psycho-oncologists and stakeholders with insights on this topic. We removed lies 101-104 accordingly

Finally, should you have space, it may be important to further introduce/explain/breakdown what CIT is.

RESPONSE: Thank you. We agreed and explained CIT further in the method section.

The methods are generally clear and straightforward. However, as mentioned above, it may be relevant to provide further explanations as to your used of CIT.

RESPONSE: Thank you. We added an explanation concerning CIT as requested.

I suggest you also provide more information on your use of triangulation.

RESPONSE: We agreed, and added information about how triangulation has been performed.

It may be worth, for reproducibility, to provide more details on how this “framework” was used/built.

RESPONSE: Thank you! We added an explanation of what framework is in this context and practically highlighted how we built it, for enhancing reproducibility.

Finally, it is unclear to me why you asked your participants about previous involvement in emergency psychology. How does that impact their answers to your questions? In what ways is this important to keep in mind? Finally, it may be important to clarify that this is an inpatient setting, right from the beginning, as the situation may be vastly different in outpatient settings.

RESPONSE: Thank you for this comment. At first, we thought that having previous experience in such a discipline could impact professionals and helped them coping with the events positively. However, we did not have any indication of this. We decided to simplify the text accordingly.

The results are well outlined, and themes appear clear and distinct. I suggest making theme 1 more explicit in the ways it relates to PTs’ practice and how they adapted. It appears to me that it is more strongly related to patients’ experience rather than PTs’ and therefore does not add much to your research question.

RESPONSE: Thank you. We explicitly related the theme 1 to the participants’ perspective.

In addition, the disclosure from patients could be further explained. How is this different than existential dread when facing potential death?

RESPONSE: We added a clarification accordingly to theme 2.

Finally, the discussion really ties it all together. I appreciated the clearer distinction between your two bigger themes: innovation and lack of recognition. I believe both have great potential but are too tangled in your results. I suggest taking this distinction which you have mastered in the discussion and applying it to the rest of the article.

RESPONSE: Thank you for this comment. We harmonized the findings with the distinction we highlighted in the discussion, by renaming the theme and the sub-theme in the results’ section.

In addition, most themes seem to be rooted in systemic issues and situational factors. Can you further discuss the recognition of psychotherapy in hospitals using a public health or systemic lens? What do those results mean, what changes are needed? You may not need to spend as much time on ICTs.

RESPONSE: Thank you for this suggestion. We agree that the topic concerning the recognition of the role of psycho-oncology (not only in hospital but also in the other healthcare setting) is also linked to health policies issues. We integrated the discussion in this subparagraph by adding some updating on the current legal framework of psycho-oncology in Italy and on its impact on the future development of the discipline

Overall, this article addresses some important issues within psycho-oncology and sheds some light on its in hospital settings. Thank you for your submission.

RESPONSE: Thank you so much for this reassuring comment.

REVIEWER 2

Thank you for the opportunity to review your manuscript. The topic you explore is highly relevant, especially given the challenges introduced by the COVID-19 pandemic in psychological support practices within oncology. Below, I provide a detailed review of your manuscript along with several suggestions to further strengthen it.

RESPONSE: Thank you for your time and dedication in reviewing our manuscript.

The manuscript is well-written and addresses a timely and important topic. The use of a mixed-methods approach is appropriate for answering your research question. However, there are a few methodological and conceptual aspects that could be expanded to improve the transparency and clarity of the work.

RESPONSE: Thank you! We changed the paragraphs you indicated accordingly.

1. Title and Abstract

-The title is clear and descriptive, but I recommend specifying the geographic context (Italy) to make the focus on Italian psychotherapists more explicit for an international audience.

RESPONSE:Thank you. We provided a new title to address this comment.

-The abstract is clear, but the methodology section could be expanded, particularly to explain the "QUAL→qual mixed-method" approach for readers unfamiliar with this terminology.

RESPONSE: We agreed. However, given the room left for the abstract, we preferred not to mention the type of mixed methods, leaving its explanation in the method section.

2. Introduction

-The introduction provides a comprehensive background on psycho-oncology and the implications of the pandemic. The reference to the history of psycho-oncology in Italy is well done.

RESPONSE: Thank you so much for this comment.

-Provide a clearer statement of the research gap early in the introduction.

RESPONSE: We agreed. We rephrased the first paragraph of the introduction accordingly.

-Better explain the rationale for focusing on the Italian setting by relating it to international contexts, which could make the study’s findings more broadly applicable.

RESPONSE: We amended the second paragraph of the introduction for explaining the rationale you suggested.

3. Methods

-The mixed-methods approach (QUAL→qual) is appropriate, but more details are needed about how the qualitative data were triangulated and what criteria were used for selecting participants for the interviews.

RESPONSE: Thank you. We added an explanation about how we implemented triangulation. Also, we added the inclusion criteria we adopted for selecting the participants.

-Convenience sampling is acknowledged as a limitation, but the manuscript would benefit from a more in-depth discussion of how this might affect the generalizability of the findings.

RESPONSE: We agreed with this comment and added a discussion within the limitation paragraph.

-Provide more information about the coding process, such as how discrepancies between coders were resolved.

RESPONSE: To address the reviewer's request, we provide further details on the coding process and how discrepancies between coders were resolved.

Clarify whether any measures were taken to ensure that the interview sample was representative of the broader survey population.

RESPONSE: We agreed. We commented on this in the limitations’ section, as the participation to the interview was voluntary.

4. Results

-The four main themes (patient relationships, clinical practice management, emotional aspects, and organizational aspects) are well supported by participant quotes, but the results section would benefit from clearer separation between survey and interview data. Reorganize the results section to explicitly separate the findings from the survey and interview phases before integrating them in the discussion.

RESPONSE: Thank you for this comment. While we agreed that a clearer separation may enhance readability, we performed data triangulation at the data analysis stage, choosing to integrate different data sources in the analysis rather than in the discussion. The way we applied triangulation is clarified in the method section.

-Some subthemes, such as “the inner world of the psychotherapists,” are more abstract and would benefit from further elaboration or concrete examples.

RESPONSE: We agreed. We further elaborated the themes and sub-themes with concrete examples. As the subtheme mentioned, we also renamed it.

5. Discussion

-The discussion reflects thoughtfully on the challenges psychotherapists faced during the pandemic and effectively ties the findings to existing literature.

RESPONSE: Thank you for this comment.

-The discussion could benefit from a more critical evaluation of the study’s limitations, especially concerning the convenience sampling and the lack of representation from psychotherapists working with pediatric patients. Expand the discussion with a more detailed analysis of how these limitations might have influenced the results and their generalizability.

RESPONSE: Thank you! We re-wrote the limitations’ section accordingly.

-The points on the importance of ICT tools and staff support programs are well made, but the discussion could offer more concrete recommendations for policy and practice, particularly concerning how psycho-oncology services should prepare for future crises.

RESPONSE: Thank you for this comment. We agreed and offered our perspective on more concrete recommendations for policy and practice in the conclusions.

6. Conclusion

-The conclusion effectively summarizes the key contributions of the study, particularly the value of ICT tools and the need for ongoing support for psycho-oncology services. Provide more actionable recommendations on how your findings can inform clinical practice or health policy. This would help bridge the gap between research and practice.

RESPONSE: Thank you. We rewrote the conclusion adding several actionable recommendations that can be proposed to enhance psycho-oncology services and their integration into healthcare systems, based on our findings. Consequently, the former paragraph 4.5 became redundant.

7. Ethical Considerations

-Ethical approval is well described, and the consent process is clear. However, it would be helpful to briefly address any challenges related to ensuring confidentiality during remote interviews, given the sensitive nature of the topic.

RESPONSE: Thank you. We added specifications on confidentiality and privacy.

8. Figures and Tables

-The figures and tables are well-organized and informative. Table 2 is particularly helpful in contextualizing the sample. Consider adding more detail to Figure 1 (Mixed-methods design process) to clarify how the different phases of the study were integrated in the analysis.

RESPONSE: We provided a figure 1 with more detail as suggested.

-Include thematic maps to better visualize the relationships between the different themes and subthemes.

RESPONSE: We added a thematic map as figure 2.

---

## [Decision Letter · Decision Letter 1]

14 Jan 2025

Psycho-oncology practice for cancer patients during the pandemic lockdown in Italy: a qualitative mixed-method study with psychotherapists

PONE-D-24-04536R1

Dear Dr. PERIN,

We’re pleased to inform you that your manuscript has been judged scientifically suitable for publication and will be formally accepted for publication once it meets all outstanding technical requirements.

Kind regards,

Jan Christopher Cwik, Ph.D.

Academic Editor

PLOS ONE

Additional Editor Comments (optional):

Reviewers' comments:

Reviewer's Responses to Questions

**Comments to the Author**

1. If the authors have adequately addressed your comments raised in a previous round of review and you feel that this manuscript is now acceptable for publication, you may indicate that here to bypass the “Comments to the Author” section, enter your conflict of interest statement in the “Confidential to Editor” section, and submit your "Accept" recommendation.

Reviewer #1: All comments have been addressed

Reviewer #2: All comments have been addressed

2. Is the manuscript technically sound, and do the data support the conclusions?

Reviewer #1: Yes

Reviewer #2: (No Response)

3. Has the statistical analysis been performed appropriately and rigorously? 

Reviewer #1: Yes

Reviewer #2: (No Response)

4. Have the authors made all data underlying the findings in their manuscript fully available?

Reviewer #1: Yes

Reviewer #2: (No Response)

5. Is the manuscript presented in an intelligible fashion and written in standard English?

Reviewer #1: Yes

Reviewer #2: (No Response)

6. Review Comments to the Author

Reviewer #1: (No Response)

Reviewer #2: Dear Authors,

Thank you for addressing the comments and suggestions provided in the previous round of reviews. I have carefully evaluated the revised manuscript and the changes you implemented.

Overall, the revisions significantly improve the manuscript, addressing most of the concerns raised in the initial review. I believe the study is now a valuable contribution to the literature on psycho-oncology and pandemic-related healthcare challenges.

Thank you for your thoughtful revisions and for engaging constructively with the feedback provided.

7. PLOS authors have the option to publish the peer review history of their article (what does this mean? ). If published, this will include your full peer review and any attached files.

**Do you want your identity to be public for this peer review?** For information about this choice, including consent withdrawal, please see our Privacy Policy .

Reviewer #1: No

Reviewer #2: No

---

## [Editor Report · Acceptance letter]

PONE-D-24-04536R1

PLOS ONE

Dear Dr. Perin,

I'm pleased to inform you that your manuscript has been deemed suitable for publication in PLOS ONE. Congratulations! Your manuscript is now being handed over to our production team.

Kind regards,

on behalf of

Dr. Jan Christopher Cwik

Academic Editor

PLOS ONE